# Next-Generation Nanomedicine Approaches for the Management of Retinal Diseases

**DOI:** 10.3390/pharmaceutics15072005

**Published:** 2023-07-22

**Authors:** Binapani Mahaling, Shermaine W. Y. Low, Sanjay Ch, Utkarsh R. Addi, Baseer Ahmad, Thomas B. Connor, Rajiv R. Mohan, Swati Biswas, Shyam S. Chaurasia

**Affiliations:** 1Ocular Immunology and Angiogenesis Lab, Department of Ophthalmology and Visual Sciences, Medical College of Wisconsin, Milwaukee, WI 53226, USA; binapanimahaling@gmail.com (B.M.); wlow@mcw.edu (S.W.Y.L.); uaddi@mcw.edu (U.R.A.); bahmad@mcw.edu (B.A.); tconnor@mcw.edu (T.B.C.); 2Nanomedicine Research Laboratory, Department of Pharmacy, Birla Institute of Technology & Science-Pilani, Hyderabad 500078, India; chsanjay54@gmail.com (S.C.); swati.biswas@hyderabad.bits-pilani.ac.in (S.B.); 3One-Health One-Medicine Ophthalmology and Vision Research Program, University of Missouri, Columbia, MO 65211, USA; mohanr@missouri.edu; 4Department of Cell Biology, Neurobiology and Anatomy, Medical College of Wisconsin, Milwaukee, WI 53226, USA

**Keywords:** nanomedicines, retina, nanoparticles, nanostructured scaffolds, nanodevices, nanodelivery

## Abstract

Retinal diseases are one of the leading causes of blindness globally. The mainstay treatments for these blinding diseases are laser photocoagulation, vitrectomy, and repeated intravitreal injections of anti-vascular endothelial growth factor (VEGF) or steroids. Unfortunately, these therapies are associated with ocular complications like inflammation, elevated intraocular pressure, retinal detachment, endophthalmitis, and vitreous hemorrhage. Recent advances in nanomedicine seek to curtail these limitations, overcoming ocular barriers by developing non-invasive or minimally invasive delivery modalities. These modalities include delivering therapeutics to specific cellular targets in the retina, providing sustained delivery of drugs to avoid repeated intravitreal injections, and acting as a scaffold for neural tissue regeneration. These next-generation nanomedicine approaches could potentially revolutionize the treatment landscape of retinal diseases. This review describes the availability and limitations of current treatment strategies and highlights insights into the advancement of future approaches using next-generation nanomedicines to manage retinal diseases.

## 1. Introduction

The retina is a multi-layered neural tissue with a complex cellular organization at the back of the eye. It senses light and converts it into electrical signals, which the brain perceives for vision. Any dysfunction or abnormality of the retina can give rise to temporary or irreversible vision loss [1]. Therefore, globally, retinal diseases significantly cause ocular morbidity and visual impairment. The prevalence of retinal diseases ranges from 5.35% to 21.02% at age 40 years and above [2]. In developed countries, retinal diseases are the most common cause of irreversible blindness. The major retinal diseases are age-related macular degeneration (AMD), diabetic retinopathy (DR), ischemic retinopathy, proliferative vitreoretinopathy (PVR), and inherited retinal disorders (IRDs) [3]. Current treatment strategies for these retinal disorders are invasive and often associated with side effects such as raised intraocular pressure, corneal and crystalline lens irregularities, vitreous hemorrhage, or damage to retinal cells.

Nanomedicine presents a new perspective, embodying nanotechnology, nanodevices, and nanomaterials for tissue repair and drug delivery for the management of retinal diseases. It utilizes materials with a 1–100 nm dimension to function at the cellular and molecular level [4]. In recent years, nanomedicine has evolved dramatically for managing ocular diseases, providing tools for tissue engineering, implants, and drug delivery. This review highlights the recent advances in nanomedicine for managing vision-impairing retinal diseases.

## 2. Retinal Diseases and Current Treatment Strategies

Retinal diseases are the leading cause of blindness and low vision worldwide, resulting in a considerable economic burden and poor quality of life [5,6]. Common retinal diseases, current treatment strategies, and their limitations are described in Table 1.

DR is a common cause of vision loss in people with prolonged diabetes due to neurovascular complications. In current treatment strategies, neurodegeneration is often overlooked. Treatment options for DR include anti-VEGF injections into the eye, topical corticosteroid eye drops, laser photocoagulation, or vitreous surgery [7]. These intravitreal injections may require repeated doses, resulting in several ocular complications. Additionally, several patients may not respond to anti-VEGF therapy [7,8]. The financial burden and poor patient compliance also limit the effective use of intravitreal injections. Laser photocoagulation may also be performed to shrink problematic blood vessels or vitrectomy surgery. However, this invasive treatment strategy may result in discomfort, and multiple procedures may be required to protect against vision loss [7]. Furthermore, laser photocoagulation is a destructive procedure that causes permanent damage to the retina while preserving functional vision and may result in reduced night vision and field of view [10].

AMD is the leading cause of central vision loss in the aged population. Macular damage can result in poor contrast, blurry central vision, or the perception of wavy lines. The Age-Related Eye Disease Study 2 (AREDS 2) found that Lutein + zeaxanthin and omega-3 fatty acid supplementation may help slow the progression of early-stage AMD. However, these supplements can interfere with other medications and should be carefully discussed with a medical professional before consumption [12]. Anti-VEGF injections and photodynamic therapy (PDT) are other options for treating late-stage AMD. However, unlike DR, anti-VEGF injections require multiple doses due to their short durability, and PDT is infrequently used alone due to its inadequacy and unpredictability [13].

Ischemic retinopathy (IR) occurs due to an inadequate blood supply to the retina. IR leads to alterations in retinal metabolic functions and can result in vision loss and blindness. Common causes of retinal ischemia include central and branch artery or vein occlusion and DR [19]. The treatment options available for ischemic retinopathies are anti-VEGF injections, laser treatment, and vitrectomy [20]. Another ischemic disease is retinopathy of prematurity (ROP), a condition caused by the underdevelopment of retinal vessels in prematurely born infants. ROP is characterized by the proliferation of abnormal fibrovascular tissue and vessels at the border of the vascularized and non-vascularized retina. ROP treatment strategies include cryotherapy, laser photocoagulation, and anti-VEGF injections [14]. Unfortunately, these treatments have been associated with undesirable complications, such as increased intraocular pressure, vitreous hemorrhage, choroidal or retinal detachment, and the formation of choroidal neovascular membranes [21,22].

PVR is the proliferative growth of contractile membranes in the vitreous or retina that may result in tractional retinal detachment. Surgical intervention is the only management option for PVR. Vitrectomy and pre-retinal membrane removal are commonly performed procedures for PVR treatment. Also, intravitreal injections of corticosteroids or anti-proliferative agents may prevent PVR [16]. However, one of the major problems after retinal surgery is the re-occurrence of retinal fibrosis. Pharmacological intervention to inhibit vitreoretinal scarring, either preceding or following retinal reattachment surgery, promises to reduce PVR and enhance the surgical success rate and visual outcomes [17]. Nevertheless, surgery remains the primary treatment option for PVR, and post-operative complications, including ocular hypertension, subretinal hemorrhage, corneal or lens opacification, and recurrent retinal detachment, may still occur.

RP is a genetic disorder characterized by the degeneration of retinal photoreceptors, resulting in night blindness and progressive loss of vision. There is currently no treatment for RP [18]. Management of RP involves the use of low vision aids to maximize existing vision and vitamin A supplementation to slow the progression of the disorder [23]. However, there are various forms of RP, with no universal treatment targeting all forms of RP. Gene therapy trials are ongoing, using modified viral vectors to target specific retinal cell types [24]. However, the dosage, efficacy, specificity, and long-term effects of this therapy are yet to be determined. Stem cell therapy has also been explored to replace photoreceptors by growing them in vitro and implanting them into the subretinal space [18]. However, stem cell therapies for RP are still in the early stages of development.

Next-generation nanomedicines are under development to overcome several limitations associated with the currently available treatment strategies. Nanomedicines have immense potential to inhibit neurodegeneration and prevent vascular complications. They can also act as scavengers for reactive oxygen species, enhance the specificity and durability of current therapeutics, or even act as a drug delivery system to deliver a sustained release of individual or combinatory drugs.

## 3. Nanomedicine Approaches for Retinal Diseases

Nanomedicine is an emerging field of medical sciences that has been used for imaging, diagnosis, biosensors, tissue engineering, and drug delivery. Nanomedicine can be classified into (i) nanoparticles, (ii) nanostructured scaffolds, (iii) nanodevices, and (iv) nanodelivery systems. The diversity of the materials used to synthesize nanomedicines allows for manipulating the shape, size, surface properties, and ultimately their use as next-generation therapeutics for retinal diseases. A significant challenge for drug absorption lies with the limited time the drug stays on the ocular surface. Many factors contribute to this problem, including rapid tear turnover, blinking, nasolacrimal drainage, and systemic absorption. As a result, the amount of bioavailable drug within the eye is <5%, and penetration into intraocular tissues is <0.001%. In the following section, current advancements in nanomedicines for the treatment of retinal diseases are described. Table 2 and Figure 1 describe the next-generation nanomedicines available for managing retinal diseases.

### 3.1. Nanoparticles

Nanoparticles range from 1–100 nm in size and can be classified as metallic or non-metallic [4]. These nanoparticles can act as therapeutic agents by targeting systems or carriers for multiple ligands such as DNA, antibodies, peptides, molecular sensors, therapeutic molecules, and probes for retinal diseases.

#### 3.1.1. Metallic Nanoparticles

Metallic nanoparticles, such as gold, silver, platinum, cerium oxide, and yttrium oxide, have been explored as therapeutic agents for retinal diseases. Other metals, such as iron, cobalt, and nickel-based nanoparticles, have also been studied for their magnetic properties to target specific cells or to overcome the blood–retinal barrier [34,35,36].

Gold nanoparticles are highly suited for treating retinal diseases, as they can pass through the anatomical and physical barriers of the eye, such as the ocular surface, cornea, conjunctiva, sclera, choroid, and blood–retinal barriers. Additionally, when applied as eye drops, they can pass through dynamic barriers such as tear fluid, lymphatic blood vessels, and conjunctival and choroidal blood vessels [25]. The administration of these nano-invasive drugs can overcome the complexities associated with repeated intravitreal injections, such as increased intraocular pressure, endophthalmitis, cataracts, retinal toxicity, inflammation, poor localization, and retinal detachment [25,87]. Furthermore, intravenous injection of 20-nm gold nanoparticles can cross the blood–retinal barrier without causing inflammation or toxicity [26,88]. Gold nanoparticles have also been reported to inhibit retinal neovascularization in a mouse model of ROP by blocking VEGF-induced auto-phosphorylation of VEGFR-2 to inhibit ERK 1/2 activation [27]. Additionally, gold nanoparticles are able to inhibit vascular permeability and VEGF-induced cell migration [28].

Silver nanoparticles can also inhibit VEGF-induced endothelial cell migration and proliferation by acting on the PI3/AKT signaling pathway [29]. Additionally, they can inhibit VEGF and IL-1β-induced vascular permeability by acting on the Src signaling pathway [30]. These findings are essential, as many chronic retinal diseases, such as DR and AMD, are associated with vascular permeability and abnormal angiogenesis. Furthermore, other metallic nanoparticles, such as platinum, cerium oxide, and yttrium oxide, display antioxidant properties and inhibit oxidative stress-induced apoptosis in retinal cells [31,32,33]. Thus, these nanoparticles may effectively regulate the progression of oxidative stress-induced cell death in macular degeneration, RP, ischemic retinopathy, retinal vascular occlusion, DR, and other blinding diseases.

Iron oxide superparamagnetic nanoparticles can magnetize stem cells and can be delivered to the inner and outer retina using a magnetic field after intravitreal or intravenous injection [34]. This method allows for targeted drug delivery of particular cell types, such as those found in the retinal pigment epithelium (RPE) [35]. Furthermore, a transient, reversible enhancement of blood–retinal barrier breakdown using magnetic nanoparticles can enhance drug bioavailability in the retina when applied systemically. Ferritin-modified iron oxide magnetic nanoparticles have been explored to enhance drug bioavailability during induced transient hyperthermia [36]. However, dosage optimization to avoid neuronal toxicity is crucial. Nevertheless, administering drugs to the retina less invasively might be a promising strategy.

#### 3.1.2. Non-Metallic Nanoparticles

Non-metallic nanoparticles, like mesoporous silica, carbon nanospheres/nanodiamonds, and calcium phosphate, have recently been explored for drug delivery due to their biocompatible and bioresorbable properties [37,89,90,91]. Mesoporous silica nanoparticles have a large surface area, uniform pore size, high pore volume, and excellent ocular biocompatibility for retinal drug delivery. Immunomodulatory drugs like tacrolimus, antiangiogenic drugs like bevacizumab, and anticancer drugs like topotecan can be encapsulated for targeting the retina using non-metallic nanoparticles [38,39,89]. Conjugating these nanoparticles with target moieties on the surface of mesoporous silica nanoparticles further enhances cellular uptake, thereby increasing the specificity and bioavailability of the drug. For example, folic acid-modified mesoporous silica nanoparticles significantly enhanced cellular uptake by retinoblastoma cells and exhibited superior anticancer efficacy [39]. Surface modification like PEGylation and pullulan modification with ultrasound can also enhance drug retention in the blood circulation and the tumor microenvironment [37,90]. Additionally, biodegradable calcium phosphate nanoparticles loaded with hypotensive agents like 7-hydroxy-2-dipropyl-aminotetralin and lisinopril showed long-term anti-glaucoma effects by reducing the intraocular pressure in experimental animal models [40,91].

### 3.2. Nanostructured Scaffolds

Neurodegeneration is a significant consequence in the progression of retinal diseases such as AMD, DR, RP, and Stargardt’s disease. Currently, no established treatment strategies exist to stop or reverse the degenerative process completely, or to reinstate regular retinal function to restore vision [92]. Cell therapy seems to be a promising strategy to overcome this problem [93]. However, the success of retinal transplantation by injecting cell suspensions has limited cell survival and lacks cellular integration [92]. Nanostructured scaffolds such as nanofibers, nanoporous gels, nanotopography, and 3D printing can support cell adhesion, survival, proliferation, migration, differentiation, and integration, and improve visual function [92,93,94,95,96].

#### 3.2.1. Nanofibers

Nanofibers have been used as scaffolds for retinal tissue engineering and as carriers for the controlled delivery of drugs, proteins, and DNA to the retina. These nanofibers can be fabricated through electrospinning, self-assembly, and phase separation [4,97]. Nanofibers fabricated from natural and synthetic polymers have been explored for achieving cellular viability, differentiation, and integration in the retina. Transplantation in the RPE is a promising strategy for the management of AMD and Stargardt’s disease. However, the AMD-affected Bruch’s membrane may not be sufficient to support the growth of transplanted RPE cells. Ultrathin electrospun 3D nanofibrous membranes from collagen type I and poly(lactic-co-glycolic) acid (PLGA), mimicking Bruch’s membrane, have been explored to grow the RPE monolayer. Interestingly, human RPE cells grown on nanofibrous membranes exhibited correct orientation, polygonal cell shape, microvilli on their apical surfaces, and tight junctions, similar to the native human RPE. Furthermore, the cells expressed RPE65 protein, which is crucial for RPE maintenance [41]. Nanofibers from natural materials like pectin-modified polyhydroxy butyrate or soy protein have similar effects on human RPE cells, ARPE-19 cells, and stem cell-derived RPE cells, demonstrating that natural matrigels can be explored for RPE tissue engineering [43,93]. Additionally, electrospun polypyrrole–graphene-modified PLGA nanofibers have shown promising results for optic nerve regeneration in vitro [42]. However, in vivo transplantation of these cells/scaffolds has yet to be performed in the retina.

#### 3.2.2. Nanotopography

Optic neuropathies, such as glaucoma and Leber’s hereditary optic neuropathy, can result in retinal ganglion cell (RGC) death. The major challenges for RGC transplantation are maintaining cell survival and controling axon orientation to target the optic nerve head. Axonal growth of retinal cells can be guided by micro/nanopattern topography on a crafted material surface. Topography plays a vital role in neurite growth and orientation. Electrospinning and 3D printing are suitable methods for generating nanotopograhy [42,44,95]. In a previous study, RGCs were seeded on randomly arranged polypyrrole–graphene-modified PLGA nanofibers to obtain neurite growth of 80–100 µm. Additionally, the authors showed that the aligned fibers encouraged neurite growth extension up to 140 µm, along with the orientation of the fibers [42]. Gallium phosphide nanowires of different geometries can also support the long-term survival of retinal cells and the elongation of neuronal outgrowth. Cells clustered on long nanowires significantly increased neurite outgrowth compared to those clustered on short nanowires [44]. These nanowires also support synapse formation significantly better than a flat surface, suggesting the importance of topography in retinal tissue engineering [44]. In another study, poly(ethylene-co-vinyl acetate) was designed with parallel straight grooves using 3D imprinting lithography [95]. Human-induced pluripotent stem cell (iPSC)-derived RGCs grown on this scaffolding displayed organized axons, which projected axially. These studies highlight the importance of topography on neurite growth and axon guidance. However, further in vivo evaluation is still required to determine its functionality and integration into the eye.

### 3.3. Nanodevices

Nanodevices have great potential for treating retinal diseases. In recent years, nanodevices have been developed for sustained drug delivery, retinal tissue engineering, and as interface materials for retinal recording, biosensors, photosensors, neutralization of pore-forming toxins, and infrared vision. The nanodevices that have been explored for retinal use include contact lenses, carbon nanotubes, nanoprobes, nanobelts, nanosponges, and nanoantennae.

Drug delivery to the retina using contact lenses is an innovative strategy that overcomes the toxicity issues associated with intravitreal injection. It also has enhanced bioavailability compared to topical eye drops, as it overcomes tear dilution, reflex blinking, and rapid fluid drainage that collectively reduces the drug’s retention time on the ocular surface. Dexamethasone–PLGA contact lenses have shown sustained delivery of the drug to the retina and successfully inhibited retinal vascular leakage induced by intravitreal injection of VEGF in a rabbit model. The safety and biocompatibility of these contact lenses were found to be suitable for four weeks [45].

Carbon nanotubes are an attractive option for retinal tissue engineering and as an interface material for retinal function. Human RGCs grown on carbon nanotubes showed enhanced growth, physiological synaptic activity, and biocompatibility [98]. In addition, vertically aligned carbon nanotubes showed increased cell viability, proliferation, neurite outgrowth, biocompatibility, and mechanical integrity [46,47]. However, a critical review of the biological mechanisms of carbon nanotubes reported the possibility for toxicity [99]. As such, further studies are required to determine if carbon nanotubes are safe for use. One intriguing use of nanoprobes is the development of an alizarin red aluminum(III) complex conjugated to graphene oxide, which was successful in detecting intracellular lysine residues. This study provided an excellent model for monitoring lysine in retinal diseases using fluorescence spectroscopy [48]. Similarly, single-crystal selenium nanobelts with a large surface area, fine photoconductivity, and biocompatibility showed immense potential for developing implantable artificial retinas or rapid photon detector/stimulator for optogenetics. However, in vivo evaluation is still required [49].

During ocular infection, microorganisms secrete pore-forming toxins that cause damage to retinal function. Biomimetic nanosponges derived from erythrocytes showed promising neutralizing effects on the cytolysin secreted by Enterococcus faecalis and pore-forming toxins produced from Staphylococcus aureus, Enterococcus faecalis, Streptococcus pneumoniae, and Bacillus cereus, encouraging the use of nanosponges as adjunctive therapies for the treatment of ocular infections [50,51]. One of the most interesting findings was the development of nanoantennae to visualize near-infrared light. The nanoantennae consisted of photoreceptor-binding upconversion nanoparticles that acted as near-infrared light transducers. Mice injected with nanoantennae could differentiate sophisticated near-infrared shape patterns and showed no potential toxicity [52]. These developments demonstrate potential therapeutic opportunities for emerging bio-integrated nanodevice designs and applications.

### 3.4. Nanostructure-Based Drug Delivery Systems

One of the most critical advances in nanomedicine is the development of nanoparticle-based drug delivery systems for retinal diseases. These newer delivery systems are anticipated to prevent the drawbacks of conventional drug administration. Nanostructure-based drug delivery systems will (i) provide a non-invasive drug delivery system that can be applied via eye drops, which cross the ocular barriers to reach the retina for enhanced bioavailability; (ii) provide targeted systemic delivery to overcome the blood–retinal barriers, thereby reducing the drug dosage and overcoming systemic barriers; (iii) increase the half-life of the drug to prevent any complexities associated with repeated intraocular injections; (iv) aid in sustained drug delivery for the long durations required to treat chronic diseases like DR and AMD; (v) enhance the versatility of the drug delivery system to allow for the delivery of hydrophilic and hydrophobic drugs, proteins, peptides, DNA or RNA, and aptamers; and (vi) help modulate stimulus-responsive drug delivery systems. In this context, the nanostructure-mediated drug delivery systems that have been explored are polymeric nanoparticles, lipid nanoparticles, liposomes, polymeric micelles, dendrimers, nanoemulsions, and nanogels.

#### 3.4.1. Polymeric Nanoparticles

Polymeric nanoparticles are used for the retinal delivery of drugs ranging from 10–1000 nm in size [4]. Polymeric nanoparticles show excellent bioavailability, biocompatibility, and biodegradability in the retina. There are different types of polymeric nanoparticles, such as nanocapsules or nanospheres, in which a drug can be entrapped/encapsulated or dissolved in a matrix. The ability of drug-loaded nanoparticles to overcome physiological barriers and redirect drugs to the site of action through passive diffusion or targeted mechanisms has become an adaptive method for researchers in designing polymeric nanoparticulate delivery systems [100,101]. Polymeric nanoparticulate drug carriers offer various advantages, such as imparting better stability than lipidic carriers, preventing entrapped drugs from degradation, releasing drugs sustainably, and allowing better penetration through biological barriers. However, despite numerous advantages, polymeric nanoparticles have limitations, such as toxicity and slow clearance [102,103].

Biodegradable polymers play an essential role in the formulation of polymeric nanoparticles, and particle size (nanoscale) is critical for ophthalmic formulations to avoid patient discomfort. After applying nanoparticles as eyedrops, the nanoparticles need to be retained in the ocular cul-de-sac to allow for adequate retention and achieve sustained drug release. Hence, designing nanoparticles with biomaterials is vital to retain the drug appropriately. Many biodegradable polymers have been developed to improve ocular drug delivery, including chitosan, hyaluronic acid (HA), PLGA, polyacrylamides, polylactic acid (PLA), poly ε-caprolactone, and albumin [104,105,106]. Chitosan, a mucoadhesive polymer, has been used to improve the retention time of nanoparticles. This improved retention time has been attributed to chitosan’s ability to develop strong molecular attractions between the negatively charged sialic acid residues of mucin and chitosan’s positive amino acid groups [107]. The various polymers utilized in the preparation of polymeric nanoparticles for the management of retinal diseases are described in Table 3.

Polymer nanoparticles can be fabricated using various methods, such as solvent evaporation, desolvation, emulsification, and ionotropic gelation. Several polymers have been utilized to deliver chemical entities, such as drugs, peptides, genes, and proteins, to the target tissues. However, selecting appropriate polymers with a good biocompatibility and biodegradability has been critical for nanoparticle design [105]. Physicochemical properties, such as size, surface charge, hydrophobicity, surface functionality, and mucoadhesive properties, can be controlled to make the nanoparticles more suitable for retinal drug delivery [116,117]. Polymeric nanoparticles with hydrophobic cores and hydrophilic mucoadhesive shells have been delivered to the retina. One example demonstrated that applying triamcinolone acetonide-loaded nanoparticles as eye drops significantly inhibited inflammation, neurodegeneration, and angiogenesis in a DR rat model [8]. Furthermore, a dual drug delivery system developed by the core–shell nanoparticle system showed superior anti-inflammatory and antibacterial properties in vitro, demonstrating a promising strategy for developing non-invasive therapies for endophthalmitis [21]. In addition, a single intravenous injection of PLGA nanoparticles modified with arginine–glycine–aspartate (RGD) peptide significantly reduced angiogenesis and fibrosis in primates and murine models of choroidal neovascularization [53]. This study provided a proof-of-concept for the targeted intravenous delivery of nanoparticles with an increased half-life. In another study, doxorubicin-conjugated polyethylene glycol and poly(sebacic acid) nanoformulation (DXR-PSA-PEG_3_) successfully delivered a drug to the eye, with detectable levels in the aqueous and vitreous humor for at least 105 days. The study also suggested significant inhibition in choroidal neovascularization for 35 days in a rabbit choroidal neovascularization (CNV) model [56]. The versatility in the application of polymeric nanoparticles for retinal diseases was further demonstrated in several studies in which PLGA nanoparticles were reported to (i) encapsulate hydrophobic drug-like celecoxib for the inhibition of oxidative stress in a DR rat model, (ii) work as hydrophilic dye-like rhodamine 123 for retinal imaging, (iii) bind protein/antibody-like bevacizumab for long-term inhibition of choroidal neovascularization, and (iv) deliver siRNA and plasmid DNA in vivo to the retina and choroid to prevent neovascularization [53,57,58,59,60]. Recent developments have led to stimulus-like ultrasound and light-responsive nanoparticles as next-generation nanomedicines for retinal diseases. Intravitreal injection of ultrasound-sensitive monomethoxypoly(ethylene glycol)-PLGA-poly-L-lysine nanoparticles has successfully delivered siRNA to the neural retina. In this study, ultrasound significantly down-regulated the mRNA and protein expression of PDGF-BB [60]. Additionally, blue light exposure cleaved the 7-(diethylamino)-4-(hydroxymethyl)-coumarin (DEACM) photocleavable group attached to the cell-penetrating peptide (CPP)-modified PEG-PLA nanoparticles delivered intravenously and caused a significant reduction in neovascular lesions in mice [118].

#### 3.4.2. Lipid Nanoparticles

Currently, the lipid nanoparticles used for retinal drug delivery vary from 10–1000 nm. Lipid nanoparticles are comprised of solid lipid nanoparticles, nanostructured lipid carriers, and lipid drug conjugates. These nanoparticles are biocompatible and biodegradable, with enhanced drug solubility and increased bioavailability in the retina. They can also be produced on a large scale [65]. Out of the three categories mentioned above, solid lipid nanoparticles are most effectively used for carrying the payload. In an aqueous dispersion, a layer of surfactants surrounds the solid lipid matrix. Solid lipid nanoparticles are mainly produced by applying solidified nanoemulsion technologies in which high-pressure homogenization, hot-melt emulsification, and ultrasonication are most widely used. Other preparation methods include solvent-based techniques, micro-emulsification, coacervation methods, and membrane hydration [119]. In ocular drug delivery, liposomes have advantages, such as achieving high encapsulation efficiency and permeation, increasing corneal permeability and retention time, controlling release of drugs, having no toxicity, and enhancing bioavailability and biodistribution. Lipid nanoparticles also maintain adequate drug levels in the posterior segment of the eye (aqueous/vitreous humor and retina). These advantages make them ideal candidates for delivering molecules to the posterior segment of the eye. Lipid nanoparticles have some disadvantages, such as an initial burst release of drugs, and reduced drug loading capacity, and the formulation can subject to polymorphic changes upon longer storage. Additionally, the toxicity of lipid nanoparticles to retinal cells is not fully understood [120,121].

Nanostructured lipid carriers (NLCs) have a solid lipid core comprising a combination of solid–liquid and liquid–liquid interfaces with unique characteristic features. The matrix obtained from the solid–lipid and liquid–lipid mixture has a lower melting point than the original solid lipid. Various types of NLCs (imperfect, amorphous, and multiple) can be prepared depending on the lipid composition and production methods. A small change in the oil can affect the lipid crystallization, changing it into an imperfect type. Upon mixing some unique lipids, the amorphous type of lipids can be obtained, wherein the lipid matrix is solid and not crystalline. In multiple types, to improve the encapsulation of the payload into the lipid matrix and prevent compound leakage during storage, the lipid’s nanostructure can be tuned by adding more oil, followed by mixing with a solid–lipid [119].

The administration of solid–lipid nanoparticles as eye drops has been reported to reach the posterior part of the eye [64,122]. Intravenous injection of solid–lipid nanoparticles loaded with tobramycin showed higher drug bioavailability in the retina compared to topical eye drops. However, it is interesting that it can cross the ocular blood barriers in the retina [64]. Moreover, solid–lipid nanoparticles are good tools for delivering genes to the retina through intravitreal or intravenous injections [66,123]. Due to their structural advantages, solid–lipid nanoparticles have been loaded with hydrophilic drugs, hydrophobic drugs, DNA, siRNA, and miRNA for retinal diseases [65,66,67,68]. Furthermore, a recent study demonstrated the successful delivery of a switchable lipid nanoparticle based on dual drug delivery of miRNA-181 and melphalan, which significantly reduced retinoblastoma in an in vivo rat model [68]. The various types of lipids, preparation methods, and their potential applications in retinal diseases are summarized in Table 4.

#### 3.4.3. Liposomes

Liposomes are phospholipids containing an aqueous core and bilayer phospholipids capable of loading hydrophilic and hydrophobic drugs. The unique arrangement of liposomal structures allows for hydrophilic drugs to be loaded into the aqueous core through passive loading methods. In contrast, hydrophobic drugs are encapsulated in a phospholipid bilayer through passive or postinsertion methods. Liposomes have been tried as eye drops for drug and gene delivery in the retina. Commonly used liposomes range from 10–10,000 nm. Liposomes, upon disintegration, can be readily metabolized due to the presence of phospholipids, and they show low toxicity and immunogenicity. Liposomes have been widely used as a potential solution for overcoming challenges in treating retinal diseases. They can be loaded with multiple therapeutics, and their surface can be functionalized for targeting several retina receptors. Liposomes also offer intracellular drug release, and they bypass the ocular barriers [133]. Their properties are greatly affected by their chemical composition and preparation methods. The hydrophobic tail length of liposomes is influenced by properties such as the thickness and rigidity of the phospholipid bilayer. Bilayer rigidity is influenced by critical factors, including encapsulation of non-phospholipid molecules, liposome diameter, and the ratio of the aqueous core to the phospholipid bilayer [134]. The major advantages of liposomes as nanocarriers lie in their ability to enhance the permeation of poorly absorbed drugs by attaching to the corneal surface and improving the corneal residence time. Other advantages of liposomes include biodegradability, biocompatibility, the ability to encapsulate both hydrophilic and hydrophobic drugs, an enhanced pharmacokinetic profile, and low toxicity at higher concentrations. Liposomes have some limitations related to stability and increased particle size upon storage, susceptibility to being engulfed by phagocytes, and limited retention time in the retina [135,136].

One studies has suggested that liposomes can be retained in the retina for 180 min [70]. A edaravone/berberine hydrochloride-loaded liposome eye drop formulation significantly protected the retina against photo-induced damage [71,72]. Successful gene transfer using liposomal eye drops has also been reported in the retina, though it has not been evaluated in retinal disease models [73]. In a previous study, intravitreal injection of a siRNA-entrapped hyaluronic acid-coated lipoplex demonstrated significant neuroprotective effects in a retinal light damage model, without evidence of retinal toxicity [137]. Systemic administration of a rosiglitazone-loaded liposomal formulation resulted in neuroprotection in the retina of a rotenone-insult Parkinson’s disease rodent model [138]. One major drawback of liposomal delivery is the retention time in the retina, which is usually less than 24 h. [137]. However, due to their structural advantage, liposomes can encapsulate hydrophobic and hydrophilic drugs, DNA, RNA, and polypeptides or proteins, making them attractive options for treating retinal diseases [71,73,137,139]. Therefore, liposomal preparation may be adequate for the short-term delivery of drugs, but unsuitable for chronic conditions where repeated administration may be needed. Recent advances in the application of liposomes in retinal diseases are summarized in Table 5.

#### 3.4.4. Polymeric Micelles

Polymeric micelles consist of self-assembled units in a nanoscale structure with a hydrophobic core and a hydrophilic corona ranging from 10–200 nm. These micelles are biodegradable and biocompatible, making them suited for retinal drug delivery. The self-assembly of the block polymers or amphiphilic molecules takes place in an aqueous environment at a concentration more than the critical micellar concentration (CMC) [147]. Therapeutic outcomes can be improved by using polymeric micelle’s prolonged retention, as they have the potential to solubilize, stabilize, and encapsulate hydrophobic compounds [148]. The premature degeneration release of therapeutic cargo from the micelles can be prevented by crosslinking their core/corona through covalent bonding, hydrogen bonding, or π–π stacking [149]. After topical administration, the polymeric micelles penetrate through the cornea due to their small size via the conjunctival–scleral pathway [150]. Polymeric micelles reach the posterior segment of the eye through lateral diffusion from the broader conjunctival surface area. After the micelles reach the posterior segment, the RPE cells, through endocytosis, can engulf them, resulting in higher therapeutic concentrations.

Nevertheless, the surface charge and particle size of micelles define their cellular uptake and tissue absorption [151]. Polymeric micelles offer high drug loading capacity, enhanced stability, small particle size, high water solubility, and tunable surface modification. They are mainly preferred to improve the delivery of hydrophobic drugs, enhancing their therapeutic efficiency. However, polymeric micelles lack adequate retention times, are susceptible to accumulation at the target site, and are difficult to produce on a large scale [152,153].

For drug delivery, most polymeric micelles comprise different types of polymers, such as amphiphilic di-block (hydrophilic-hydrophobic) polymers, tri-block (hydrophilic-hydrophobic-hydrophilic) polymers, graft, and ionic (hydrophilic-ionic) co-polymers. Hydrophilic polymers that are primarily used in the preparation of polymeric micelles include polyethylene glycol (PEG), hydroxy propyl methyl acrylamide (HPMA), polyethylene oxide (PEO), and poly(vinyl pyrrolidine). The hydrophobic core of the polymeric micelles can be comprised of poly(L-amino acids) such as poly(L-aspartate) and poly(L-glutamate), polyesters such as polyglycolic acid and poly(D-lactic acid), and co-polymers such as PLGA and polycaprolactone [152] Polymeric micelles can be prepared using various methods such as direct dissolution, dialysis, oil/water emulsion, thin-film hydration, solvent evaporation, and freeze-drying methods [152].

The administration of polymeric nanomicellar eye drops containing rapamycin has been shown to increase the availability of rapamycin in the rabbit retina and choroidal tissues, suggesting that the sequestration of rapamycin in lipoidal retinal tissues via a non-corneal route (conjunctiva–sclera–choroid–retina) might be the preferred method for delivery of micellar drugs to the retina [74]. In another study, intravenous administration of polymeric nanomicelles encapsulating plasmid DNA of yellow fluorescent protein (pYFP) was found to be localized in the choroidal vascularization area in a CNV mouse model. Additionally, CNV was significantly reduced in psFlt-1-encapsulated polymeric nanomicelles compared to the control group. These results suggest the usefulness of polymeric micelles for non-viral gene therapy for CNV [75]. Additionally, when hexadecyloxypropyl-cidofovir micelle formulation was administered via intravitreal injection in a rabbit retinitis model, the therapeutic level of the drug was detected for up to nine weeks. Micellar formulation halted active HSV-1 retinitis in 80% of the infected eyes. In contrast, the non-micellar formulation failed to suppress active HSV-1 retinitis, indicating that lipid-derived nucleoside analogs could be formulated as micelles for intravitreal injection for sustained release of drugs in chronic retinal diseases [76]. As polymeric micelles encapsulate hydrophobic drugs more efficiently due to their hydrophobic core and hydrophilic shell, they have also been explored for siRNA delivery to the RPE and gene therapy for the retina [79].

#### 3.4.5. Dendrimers

Dendrimers are highly branched, spherical dendritic nanometers ranging from 3 to 20 nm. Polyamidoamine (PAMAM) dendrimers have mostly been explored for their water-soluble and non-immunogenic properties. The biocompatibility of PAMAM can also be increased through PEGylation [61]. Furthermore, penetrating modified PEGylated PAMAM dendrimer eye drops result in greater distribution throughout the cornea and the retina. Further modification can extend retention times for more than 12 h in the retina [61]. In an in vitro study, the PEGylated PAMAM dendrimer has been explored for siRNA delivery to retinal microvascular endothelial cells to inhibit angiogenesis. However, in vivo, its efficacy has not been evaluated [154]. Another study showed systemic and intravitreal administration of fluorescently labeled dendrimers selectively co-localized with activated microglia and astrocytes for 21 days in the retina. However, the study also suggested that a thirty-times higher dosage was required for the intravenous administration of dendrimers compared to intravitreous administration to reach the retina [62]. Furthermore, intravitreal injection of a dendrimer drug conjugate significantly inhibited neuroinflammation in a rat model of retinal degeneration [63]. Intraocular injection of anti-VEGF oligonucleotide conjugate has also been reported to significantly alleviate CNV for four months compared to the control group without toxicity, indicating the long-term safety of dendrimers in retinal neovascular diseases [54]. In comparison to other drug delivery systems, dendrimers have some unique advantages, such as a controllable structure and size, the potential for more surface modifications, the feasibility of conjugating drugs and other therapeutic molecules to the surface of active terminal groups, a lower molecular volume compared to the other linear polymers of similar molecular weights, and controlled drug release with effective targeting. Despite these advantages, dendrimers have some limitations, including a high cost for synthesis, elimination, and metabolism depending on their generation and materials used. Additionally, their cellular toxicity has not been adequately assessed [155,156].

#### 3.4.6. Nanoemulsions

Nanoemulsions are nanometer droplets made through the heterogeneous dispersions of two immiscible liquids (oil-in-water or water-in-oil), ranging from 10–1000 nm. Nanoemulsions are biocompatible, with adequate ocular tolerance [80]. Moreover, nanoemulsions can solubilize hydrophobic and hydrophilic drugs to enhance drug stability [4]. Previous studies have suggested that the administration of nanoemulsions as eye drops can follow corneal and scleral pathways to the retina [80,157]. In another study, fenofibrate-loaded nanoemulsion eye drops reduced retinal inflammation and vascular leakage in rodent models of DR and laser-induced CNV [81]. Furthermore, when applied as eye drops or through intravitreal injection, cationic nanoemulsions containing antisense oligonucleotides showed a significant reduction in neovascularization compared to the untreated control group. Cationic nanoemulsion can thus be considered a promising potential therapeutic tool for the management of retinal neovascularization [82]. Nanoemulsions have a high surface area and a small droplet size, enhancing drug absorption. They are also physically stable, non-irritating, and non-toxic to the eye. The major drawback of nanoemulsions is the requirement for many surfactants, and they have a low stability in acidic conditions. Nanoemulsions with particle sizes of more than 100 nm are known to cause blurred vision due to changes in the formulation properties. Due to the use of highly concentrated surfactants, the formulation has reduced ocular tolerance [158,159].

#### 3.4.7. Nanogels

Nanogels are cross-linked polymer networks ranging from 1–1000 nm. Nanogels are biocompatible, flexible to design, stable in aqueous systems, and are responsive to external stimuli, with a high drug loading capacity and controlled release [84]. However, there are few reports using nanogels for drug delivery to the retina. In one study, a penetration-complexed redox-responsive hyaluronic acid-based nanogel was used to release and deliver a drug (visual chromophore analog, 9-cis-retinal) to the posterior part of the eye via eye drops. Interestingly, the nanogels released the drug in a reducing environment, similar to the cytoplasm. The nanogel treatment resulted in partial recovery of photoreceptor function in treated eyes compared to the untreated controls [85]. In an in vitro study, a chitosan-based nanogel was significantly internalized by human retinal pigment epithelial cells (ARPE-19), cells with no cytotoxic or inflammatory response. However, in vivo studies on nanogels targeting the RPE have not been reported. In another study, acetazolamide-loaded nanovesicles embedded in chitosan–sodium tripolyphosphate nanogels significantly reduced ocular pressure in rabbits compared to oral tablets. Furthermore, the nanogel system was suggested to result in decreased irritation compared to free acetazolamide, indicating that nanogels can be an attractive strategy for treating ocular diseases [86].

Nanogels are known to prolong retention times in the eye and enhance bioavailability. Nanogels consist of long polymer chains containing many hydrophilic groups such as -OH, -COOH, and -CONH_2_. Due to the presence of these groups, nanogels absorb water and retain large amounts of fluids. Nanogels are highly biocompatible and protect the payload from degradation and elimination. In addition to these properties, nanogels offer a stimulus-responsive release of the payload. Nanogel bases are prepared using semi-solids, and due to the presence of refractive indices in tears, they can cause blurred vision and result in imprecise dosage [160,161].

## 4. The Future of Nanomedicines in the Management of Retinal Diseases

The challenges and exciting future of nanomedicines for managing retinal diseases are discussed below.

### 4.1. Overcoming Blood–Ocular Barriers

Developing therapeutic nanoparticles incorporated into eye drops is one of the most crucial challenges for overcoming the invasiveness of current therapeutic strategies. The challenges for treating chronic retinal diseases with nanomedicines will be creating nanoparticles that can cross static and dynamic blood–ocular barriers and provide therapeutic effects for sustained durations. Designing appropriate non-invasive or minimally invasive drug delivery systems for the retina and controlling the physicochemical properties of delivery systems are essential [116,117]. Furthermore, integrating hydrogels with polymeric nanoparticles, liposomes, micelles, dendrimers, and cyclo-dextrins can aid in drug delivery and release [162,163,164,165]. Pharmacokinetics and drug loading capacity can also be controlled by optimizing the size and concentration of the loaded nanomedicines. Additionally, cross-linking of hydrogels can alter pharmacokinetics, allowing for fine-tuning and optimization of drug release to the retina.

### 4.2. Cell-Specific Delivery in the Retina

Cell-specific targeted delivery of therapeutics to the retina is desirable. Developing targeted drug delivery systems to activate microglia in the retina may be advantageous for inhibiting gliosis and neuroinflammation. In a study examining PLGA nanoparticles conjugated with CD47, a blood–brain barrier peptide loaded with the microglia-modulating agent Nec-1s specifically targeted microglia and converted them from a pathological to a beneficial state [166]. Another study on glucocorticoid-loaded inorganic–organic hybrid nanoparticles found that they were preferentially engulfed by macrophages. The study showed promising results in preventing neuroinflammatory diseases [167], suggesting the benefits of the targeted delivery of nanomedicines.

Astrocytes are macroglia that contribute to neurodegenerative retinal diseases [168]. Interestingly, astrocytes can be targeted by dendrimers in the brain [169]. This opens new opportunities for similar technology to be applied to retinal astrocytes, which can be further extrapolated for treating neurovascular complications in the retina. Müller glia are another macroglia cell type that plays a critical role in the regulation of retinal homeostasis. Activation of Müller glia has been reported in several chronic retinal diseases. Targeting Müller glia using lipid nanoparticles has shown some promise in regulating retinal homeostasis [170]. Furthermore, targeted delivery of overexpressed adhesion molecules in retinal endothelial cells may aid in managing retinal vascular diseases [171]. Several reports have discussed RPE-targeting nanoparticles [35,172,173], encouraging future treatments against AMD and other RPE-associated retinal diseases. Additionally, intravitreal injection of PAMAM–PVL–PEG unimolecular micelle nanoparticles has been reported to accumulate in RGCs and has been shown to possess sustained viability for 14 days [174]. This shines promising light on the possibility of developing effective drug delivery systems for retinal diseases. Lastly, compact DNA nanoparticles have shown significant gene transfer abilities in photoreceptors after subretinal injection, setting the premise for potential treatments for rescuing photoreceptor degeneration [175].

### 4.3. Safety and Efficacy

Successful nanomedicines for the retina should be sterile, scalable, efficient, and safe. Sterilization and scaleup are significant problems faced in translating drug delivery systems. The critical parameters for successful translation are the size, shape, morphology, targetability, and functionality of the developed nanomedicines. Though methods such as membrane extrusion, supercritical fluid technology, and microfluidizer technology can scale up production, the large-scale generation of functional nanoparticles is still problematic [176]. Furthermore, producing therapeutic agents that remain biologically active after their formulation is still challenging, especially for protein/polypeptide-based therapeutic agents, where structural confirmation is crucial for their function. Nanomedicine architecture is thus a crucial factor during development. Additionally, it is essential to reduce toxicity and enhance efficacy.

Advanced nanotechnology and tissue engineering should be employed to develop drug delivery systems with long-term stability and low toxicity to eliminate the need for repeated applications. Extensive, detailed, and long-term investigations should focus on the biochemical, immunological, and ophthalmological aspects of ocular toxicity and biocompatibility. Controlled in vitro and in vivo toxicological evaluation and human clinical trials should also be conducted before entry into the market.

### 4.4. Drug Delivery Systems

Drug delivery systems activated via non-invasive stimuli are in the early stages of development. For example, pH and ultrasound-responsive lipid nanoparticles have been reported as effective treatments for RPE and cancer cells [69,177]. However, in vivo evaluation of these nanoparticles in the retina is warranted for further validation. In addition, different pH and ultrasound-responsive liposomes, micelles, and nanoemulsions reported for cancer have not yet been explored for retinal diseases [77,78,83,178]. Testing these responsive nanoparticles in the eye may provide new insights into their viability as a therapeutic agent. Reactive oxygen species (ROS)-responsive dendrimers might also be an attractive option for managing retinal diseases such as DR, AMD, and ROP [55]. Additionally, since the eye is sensitive to different light sources, photo-responsive nanomedicines may be a promising future drug delivery system that can be easily controlled. A recent study examining nanoparticles with peptides that contain photocleavable bonds demonstrated favorable effects in a laser CNV model [118]. Further development in stimulus-induced drug delivery systems could be the next generation of nanomedicines.

### 4.5. Gene Therapy

Gene therapy has evolved rapidly, with various non-viral vectors being explored for their cytocompatibility and reduced immunogenicity [179]. Non-viral vectors that form nanoparticle complexes can deliver genes into specific cell systems. One such example is the use of polyethyleneimine (PEI). PEI is efficient in gene transfer, without affecting corneal cell viability or morphology. Furthermore, the delivery of decorin gene therapy using PEI nanoparticles in our previous study reduced TGFβ-induced fibrosis in an in vitro model of equine corneal fibrosis [180]. This approach can open new doors for delivering gene therapies or other therapeutic agents into different parts of the eye [37,181]. Our recent study reported that polyethylene amine-conjugated gold nanoparticles showed substantial BMP7 gene delivery into rabbit keratocytes in vivo. Additionally, it inhibited fibrosis, suggesting its versatility in therapeutic applications [182], which can be further explored for treating retinal fibrosis.

### 4.6. Precision Nanomedicine

Precision nanomedicine will be the next step forward for the therapeutic development of drugs. Retinal diseases such as DR, IR, ROP, AMD, and PVR increase vascular permeability [183,184]. Delivery systems that detect fluorescent nanoparticles can be designed to distinguish the permeation effect in the retina for diagnostic purposes, as reported in cancer treatments [185]. Precision nanomedicine can be designed based on the degree of permeability by controlling drug loading, release, and pharmacokinetics. For example, the amount of drug can be controlled in a nanowafer, an ultra-thin transparent lens prepared by filling drug solutions into the dissoluble PVA template and adjusting the well size [186]. Although precision nanomedicine is still in early stages of development, future nanotools and nanotechniques will better design therapeutic drugs and molecules to manage retinal diseases.

## 5. Conclusions

Over the past decade, nanomedicine research has made several potential advancements in diagnosing, monitoring, and managing retinal diseases. In addition, recent studies have suggested that nanomedicines may act as next-generation therapeutic agents for neovascular complications, enhance the half-life of therapeutic agents, provide an alternative for the sustained delivery of therapeutic drugs, and act as a scaffold for tissue engineering in the retina. Nanomedicine has immense potential to revolutionize the current treatment strategies by overcoming the limitations associated with invasive therapies, providing targeted drug delivery, offering sustained-release formulations, and promoting tissue regeneration for managing retinal diseases. Overall, nanomedicine holds promise for improving patient outcomes and quality of life.

## Figures and Tables

**Figure 1 pharmaceutics-15-02005-f001:**
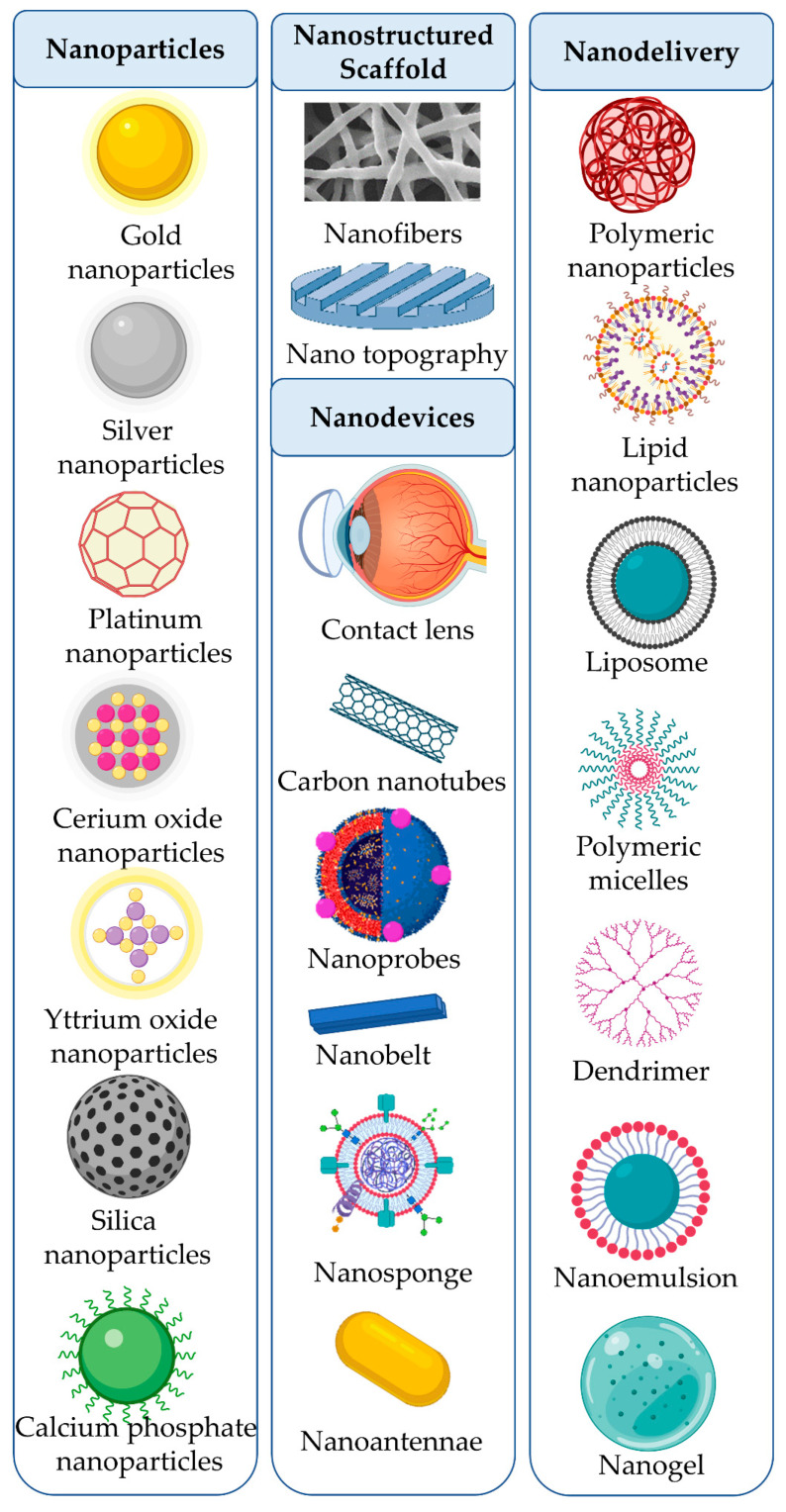
Schematic diagram depicting next-generation nanomedicine for the management of retinal diseases.

**Table 1 pharmaceutics-15-02005-t001:** Retinal diseases, current treatment strategies, and their limitations.

Retinal Diseases	Current Treatment Strategies	Limitations
Diabetic retinopathy (DR) and ischemic retinopathy (IR)	Intravitreal injection of anti-VEGF *	Complexities and treatment burden (monthly visits) associated with multiple intravitreal injections and treatment non-responders [7,8,9]
	Laser photocoagulation	Causes permanent damage to the retinal cells, can leave scotomas/blind spots [7,10]
	Vitrectomy	Surgical and anesthesia risks, post-operative infection/inflammation or retinal detachment [7,11]
Age-related macular degeneration (AMD)	Lutein + zeaxanthin	Daily administration may lead to adherence problems, cost can be a concern [12]
	Intravitreal injection of anti-VEGF	Risk and treatment burden associated with multiple intravitreal injections [9]
	Photodynamic therapy	Limited efficacy [13]
Retinopathy of prematurity (ROP) and proliferative vitreoretinopathy (PVR)	Cryotherapy and laser photocoagulation	Cornea, iris, and lens burns, hyphema [14]
	Intravitreal injection of anti-VEGF	Complexities associated with multiple intravitreal injections and retinal detachment [9,14,15]
	Vitrectomy	Vitreous or subretinal hemorrhage [16]
Retinopathy of prematurity (ROP) and inherited retinal diseases (IRDs)	Pre-retinal membrane removal or retinal reattachment surgery	Reoccurrence of retinal fibrosis [17]
	No treatment available, gene therapy and stem cell therapy are under investigation	Issues related to viral gene therapy. Direct administration of stem cells may have difficulty with integration [18]

* VEGF—Vascular endothelial growth factor.

**Table 2 pharmaceutics-15-02005-t002:** Next-generation nanomedicines and their potential applications in retinal diseases.

Nanomedicines	Applications	Refs.
Gold nanoparticles	Delivers across ocular barriers to the retina when applied as eye drops or intravenous injections. Inhibits vascular permeability and angiogenesis in retinal diseases.	[25,26,27,28]
Silver nanoparticles	Inhibits vascular permeability, endothelial cell proliferation, and migration.	[29,30]
Platinum, cerium oxide and yttrium oxide nanoparticles	Inhibits oxidative stress-induced apoptosis in retinal cells.	[31,32,33]
Iron oxide nanoparticles	Assists stem cells and drug delivery. Provides transient BRB * breakdown for enhanced bioavailability.	[34,35,36]
Silica nanoparticles, carbon nanospheres, calcium phosphate nanoparticles	Sustained delivery of hydrophilic and hydrophobic drugs. Helps in targeted drug delivery to specific cells in the retina. Enhances bioavailability of the drug.	[37,38,39,40]
Nanofibers	Controlled delivery of drugs, proteins, and DNA to the retina. Bruch’s membrane mimic and RPE tissue engineering. Optic nerve regeneration.	[4,41,42,43]
Nano topography	Nutrient growth and orientation.	[42,44]
Contact lens	Sustained non-invasive drug delivery of the drug. Overcomes tear fluid and tear film barrier.	[45]
Carbon nanotubes	Retinal tissue engineering and interface material for retinal function. Supports cell viability, proliferation, neurite outgrowth, neuron biocompatibility, and mechanical integrity.	[46,47]
Nanoprobes	Detection of intracellular molecules in retina.	[48]
Selenium nanobelts	Implantable artificial retina or rapid photon detector/stimulator for optogenetics.	[49]
Nanosponge	Neutralizes toxins and acts as adjunctive therapy for the treatment of ocular infections.	[50,51]
Nanoantennae	Near-infrared light transducer and night vision.	[52]
Polymeric nanoparticles Dendrimers	Non-invasive drug delivery system as eye drops. Targets systemic delivery. Increases half-life of the drug. Sustained delivery for months. The versatility of the drug delivery system will allow the delivery of hydrophilic and hydrophobic drugs, proteins or peptides, DNA or RNA, and aptamers. Stimulus-responsive drug delivery systems.	[8,53,54,55,56,57,58,59,60,61,62,63]
Lipid nanoparticles Liposomes	Non-invasive drug delivery system as eye drops. Targets systemic delivery. Increases half-life of the drug. Sustained delivery for hours. The versatility of the drug delivery system will allow for the delivery of hydrophilic and hydrophobic drugs, proteins or peptides, DNA or RNA, and aptamers. Stimulus-responsive drug delivery systems.	[64,65,66,67,68,69,70,71,72,73]
Polymeric micelles	Non-invasive drug delivery system as eye drops. Targets systemic delivery. Increases half-life of the drug. Sustained delivery for days. The versatility of the drug delivery system allows for the delivery of hydrophobic drugs, DNA, or RNA. Stimulus-responsive drug delivery systems.	[74,75,76,77,78,79]
Nanoemulsion Nanogels	Non-invasive drug delivery system which can be applied as eye drops. Increase in half-life of the drug. Sustained delivery for days. Stimulus-responsive drug delivery systems.	[80,81,82,83,84,85,86]

* BRB—Blood–retinal barrier.

**Table 3 pharmaceutics-15-02005-t003:** Polymers used for preparing polymeric nanoparticles and their potential applications in retinal diseases.

Polymers	Method of the Preparation	Characterization	Applications	Refs.
N-isopropylacrylamide and acrylamide	Radical polymerization	A preventive treatment for retinal degeneration using neuroprotective polymeric nanoparticulate systems was explored. The nanoparticles showed a size distribution of 22.81 ± 3.17 nm.	The formulation was injected intravitreally into the left eye of zebrafish embryos. The formulation increased the residence time over the cornea, resulting in a neuroprotective response against oxidative stress in both in vitro and in vivo studies.	[108]
PLGA-PEG	Double emulsion solvent evaporation	Memantine-loaded PLGA-PEG nanoparticles were formulated to prevent retinal ganglion cell (RGC) loss. The drug-loaded nanoparticles showed a mean particle diameter of <200 nm.	The formulation was used as a topical administration (eye drops) in a rodent model. The formulation was neuroprotective in an ocular hypertension rodent model by significantly conserving the RGC density.	[109]
Chitosan-coated PLGA	Oil-in-water emulsion	A topical formulation was developed by loading triamcinolone acetonide into chitosan-coated PLGA nanoparticles for treating retinal vasculopathy. Nanoparticles were obtained with a particle diameter of 334 ± 67 to 386 ± 15 nm.	The formulation was developed as a topical administration. The formulation showed controlled drug release and no toxicity to the ocular surface.	[110]
Chitosan functionalized PLGA	Nanoprecipitation	Sirolimus drug was loaded into nanoparticles with a mean particle size of 265.9 ± 6.3 nm.	The Sirolimus-loaded nanoparticles were administered via subconjunctival injection. Over 21 days, sirolimus-loaded nanoparticles prevented apoptosis in the damaged retina.	[111]
Bovine serum albumin, hyaluronic acid.	Desolvation	Apatinib-loaded nanoparticles were topically administered to treat diabetic retinopathy. The drug-loaded nanoparticles had a mean size of 222.2 ± 3.56 nm.	The formulation was developed as a topical administration, and efficacy was evaluated in a diabetic rat model. The retinal thickness significantly declined after nanoparticle treatment compared to the control group in the diabetic retinopathy-induced rat model.	[112]
PLGA-hyaluronic acid	Solvent displacement	Itraconazole-loaded PLGA-HA nanoparticles were conjugated with R5K peptide for treating AMD. The drug-loaded nanoparticles displayed a particle size of 167.5 to 217.0 nm.	The formulation was developed as an intravitreal injection to improve the bioavailability of anti-VEGF drugs. Conjugating the R5K peptide to the drug-loaded PLGA-HA significantly improved the binding affinity of VEGF molecules. With the synergetic effect of Itraconazole, the nanoparticles enhanced the treatment efficacy.	[113]
Polydopamine	Classical Stober method	Brimonidine-loaded polydopamine nanoparticles were prepared to decrease optic nerve crush-induced RGC apoptosis. The polydopamine nanoparticles showed a hydrodynamic size of 215 ± 2.6 nm.	The formulation was injected intravitreally into the mice using a Hamilton syringe. A single dose of nanoparticles via intravitreal injection was able to eliminate ROS in the retina and improve neurodegeneration.	[114]
Thiolated and methylated chitosan carboxymethyl dextran	Coacervation	A novel delivery system was developed to deliver drugs to the posterior eye segment in a retinoblastoma-induced rat model. The formulations displayed a particle size of 34–42 nm.	The formulation was injected intravitreally into the eyes of rats with retinoblastoma. The nanoparticles were able to reach the retina through diffusion after intravitreal injection.	[115]

**Table 4 pharmaceutics-15-02005-t004:** Lipid nanoparticle preparation methods and their potential applications in retina diseases.

Lipid Nanoparticle	Method of the Preparation	Characterization	Applications	Refs.
Dioctadecyl dimethyl ammonium bromide, glyceryl monostearate, soy phosphatidylcholine, 1,2-distearoyl-sn-glycero-3-phosphoethanolamine (DSPE)- PEG_2000_-maleimide	Film-ultrasonic	Solid lipid nanoparticles (SLNs) were prepared and loaded with miRNA-150 and quercetin for treating AMD. Both miRNA and quercetin-loaded SLNs exhibited a particle size between 115 to 209 nm.	The formulation was developed as an intravitreal injection and injected in a mouse model. The developed formulation suppressed the choroidal angiogenesis effectively. Due to the synergetic effect of both the therapeutic agents, the SLNs were able to lower the choroidal neovascularization.	[124]
1,2-dipalmitoyl-sn-glycero-3-phosphocholine (DPPC), hydrogenated soy phosphocholine, methoxy PEG_2000_-DSPE	Thin-film hydration	Lipopolyplexes were prepared by co-formulating polyethyleneimine and lipid nanoparticles to treat diabetic retinopathy. The lipopolyplexes were loaded with human antigen R (HuR), a VEGF-regulating protein. The formulation showed an average particle diameter of 200–250 nm.	The formulation was injected into Wistar rats intravitreally. After treating the streptozotocin-induced diabetic rats, the formulation significantly lowered the retinal HuR and VEGF levels.	[125]
Chitosan, glyceryl monostearate, linoleoyl polyoxyl-6-glycerides (Labrafil M 2125 CS)	Melt emulsification–ultrasonication	5-fluorouracil (5-FU) was loaded into the chitosan-modified lipid nanoparticles to treat diabetic retinopathy. The developed lipid nanocarriers displayed particle sizes around 163.2 ± 2.3 nm.	The formulation was developed as a non-invasive topical formulation. The formulation showed controlled release of 5-FU, and streptozotocin-induced diabetic rats showed significant therapeutic efficacy.	[126]
Medium-chain triglycerides, dioctadecyl-3,3,3,3 tetramethylindodicarbocyanine (DiO), polyethylene glycol (15)-hydroxy stearate, hydrogenated phosphatidylcholine	Phase inversion	CN03, a cyclic guanosine-3,5-monophosphate analog, was delivered using lipid nanocarriers for treating inherited retinal degenerations. The prepared lipid nanoparticles displayed a particle size of approximately 72 nm.	The formulation efficacy was tested via both topical and intravitreal administration. The lipid nanocarriers increased permeability, and the CN03 formulation protected the rd1 mouse photoreceptors when tested in retinal explants, making them ideal for treating inherited retinal degenerations.	[127]
Poloxamers 407/188, glycerol tripalmitate (GTP), soybean lecithin, stearic acid, PLGA	Emulsification	CN03-loaded lipid nanoparticles were prepared using w/o/w emulsification for treating posterior retinal diseases. The prepared CN03-loaded nanoparticles displayed a particle size of approximately 200–250 nm with good encapsulation efficiency.	The formulation was developed for use via intravitreal delivery. The formulation demonstrated excellent cellular uptake in retinal cell lines and released the loaded cargo inside the cells.	[128]
Glyceryl di behenate (Compritol 888), behenoyl polyoxyl-8 glycerides (Compritol HD5 ATO)	Microemulsion	Bbetulinic acid derivative (H3, H5, and H7)-loaded lipid nanoparticles were prepared to protect the Müller cells from oxidative stress injury and maintain hemostatic functions in the retina.	The formulation was designed for systemic administration. MIO-M1 cells treated with the H5-loaded lipid nanoparticles reduced glutamate-induced ROS formation and related cell death.	[129]
Di Stearoyl phosphatidylcholine (DSPC), sterols, 1,2-dimyristoyl-rac-glycero-3-methoxy polyethylene glycol (DMG-PEG)	Rapid microfluidic mixing	Gene delivery for treating inherited retinal degenerations was designed by loading mCherry mRNA in lipid nanoparticles. The mRNA-loaded lipid nanoparticles exhibited a particle size below 80 nm, with more than 90% high encapsulation.	The formulation was administered in mice via subretinal injections. The study demonstrated the feasibility of delivering genes to the photoreceptors and retinal pigment epithelium.	[130]
Lauroyl PEG -32 glycerides, propylene glycol monocaprylate, hydrogenated coco glycerides	Hot-melt emulsification and ultrasonication	Diosmin-loaded lipid nanoparticles were prepared to treat diabetic retinopathy. The critical parameters of the nanoparticles, such as particle size and PDI, were identified and optimized using the Box–Behnken design. The optimized formulation had a particle size of approximately 85 nm, with a high encapsulation efficiency of 99%.	A topical delivery of Diosmin-loaded nanoparticles was designed and developed to treat DR. The compatibility of the formulation was tested on a human retinal pigment epithelial cell line (ARPE-19). The free drug, Diosmin, showed reduced cell viability; the highest concentration resulted in 54% of cell death, and the formulations showed no significant reduction of cell viability.	[131]
DSPE-PEG, PLGA, soybean lecithin	Membrane hydration and high-power ultrasonic method	The formulations showed a mean particle size ranging from 99–127 nm. The surface charge was −32.0 ± 1.1 mV for blank formulation. After modification with peptides, the surface charge became positive. The drug loading efficiency was 1.03% with the membrane hydration method, whereas the drug loading significantly increased with the ultrasonic method.	A non-invasive drug delivery system was developed for the delivery of Axitinib. This study uses non-invasive delivery of Axitinib-loaded lipid nanoparticles for treating oxygen-induced retinal neovascularization in the posterior cavity and laser-induced AMD. A transmembrane peptide (PENE) was added to the nanoparticles to increase the transmembrane delivery. The nanomedicine effectively inhibited neovascularization due to improved tissue penetration of the nanoparticles.	[132]

**Table 5 pharmaceutics-15-02005-t005:** Liposome preparation methods and potential applications in retinal diseases.

Liposomes	Method of the Preparation	Characterization	Applications	Refs.
1-palmitoyl-2-oleoyl-sn-glycero-3-phosphocoline (POPC), DSPE-mPEG_2000_, DSPE-N-[maleimide (PEG)_2000_], cholesterol, monocarboxylate	Thin-film hydration	Monocarboxylate-conjugated liposomes were prepared to achieve high drug uptake of CN04 in the retina for treating retinal degeneration. The liposomes had a particle size of approximately 142–145 nm, with an encapsulation efficiency of more than 75%.	The formulation was injected intravitreally into the mouse model. The pyruvate-conjugated liposomes significantly enhanced uptake up to 4-fold in HEK293T cells due to the monocarboxylate transporter conjugated to liposomes. The CN04-loaded liposomes reduced cell death in the *rd1* murine retinal degeneration model.	[140]
DSPE-PEG_2000_-NHS, DSPE-PEG_2000_-TAT, egg phosphatidylcholine, cholesterol	Thin-film evaporation	Intelligent liposomes were developed to efficiently deliver ellagic acid and oxygen into the retinopathy tissue. The liposomes were functionalized with TAT/isoDGR peptides to protect the retinal cells from damage caused by high glucose induction.	The formulation was developed to co-deliver ellagic acid and oxygen via intravenous administration or as eye drops. The liposomes blocked the VEGF-p-VEGFR2 signaling pathway. In a DR mouse model, after topically administering the drug-loaded liposomes, downregulation of GFAP and HIF-1α was observed, along with the elimination of ROS in the retina.	[141]
1,2-oleoyl-3-trimethylammonium-propane (DOTAP), 1,2-dioleoyl-sn-glycero-3-phosphocholine (DOPC), cholesterol, LipoTrust Ex Oligo (short oligonucleotides)	Thin-film evaporation	SiRNA-loaded cytoplasm-responsive stearylated-peptide was added to the surface of liposomes and prepared to treat retinal diseases such as AMD. The siRNA-loaded liposomes displayed a particle size of approximately 70–80 nm, with an encapsulation efficiency of approximately 80–100%.	A non-invasive delivery was developed for siRNA delivery and administered to the rat eye as eyedrops. The intracellular uptake of liposomes was enhanced significantly due to peptides on the liposomal surface, and significant suppression of VEGF expression was observed in rat retinal pigment epithelial cells.	[142]
DSPC, DOPC, DSPE-PEG_2000_, 1,2-distearoyl-sn-glycero-3-phosphoglycerol (DSPG)	Thin-film hydration	A liposomal formulation loaded with sunitinib was developed to treat choroidal neovascularization (CNV) by blocking the signalling pathway through inhibition of the tyrosine kinase of VEGF receptors. The liposomal formulations had a mean particle size of 104 nm, and sunitinib was entrapped efficiently up to 95%.	The liposomal formulation was administered intravitreally to the mice. After intravitreal injection, drug-loaded liposomes reduced vascular leakage in the CNV mouse model.	[143]
Soybean phosphatidylcholine, diacetyl phosphate, cholesterol, chitosan	Hydration	Chitosan-coated liposomes were prepared and loaded with triamcinolone acetonide (TA) to improve the prognosis of macular edema (ME) in retinal diseases. The drug-loaded liposomes had a mean particle size of approximately 100–105 nm, and the drug was entrapped with an encapsulation efficiency of 98%.	TA-loaded chitosan-loaded liposomes were administered to rats as eye drops. The formulation was topically administered as eye drops, and significant therapeutic efficacy in reducing retinal edema was observed.	[144]
DSPC,DSPE-PEG_2000_, DSPE-PEG-Mal, cholesterol	Reverse-phase evaporation technique	Transferrin-conjugated Ganciclovir-loaded liposomes were prepared for cytomegalovirus (CMV) retinitis treatment. The optimized formulation has a particle size lower than 100 nm.	The drug-loaded liposomal formulation was developed as an intravitreal injection and topical instillation. Human retinal pigment epithelial (ARPE-19) cells demonstrated good uptake of liposomes via transferrin receptors and inhibited the CMV-infected cells.	[145]
DSPE-PEG_2000_, HSPC (hydrogenated soy phosphatidylcholine), DMPC, DOTAP	Thin-film hydration	Ranibizumab-labeled AlexaFluor555-conjugated liposomes was synthesized to pass through the vitreous humor and adhere to the retina. The lipophilic payload-loaded liposomes had particle sizes of approximately 110–135 nm.	The liposomal formulation was administered intravitreally to mice. The findings revealed that the liposomes spread well without premature clearance in the vitreous humor and interacted effectively with the inner retinal layers.	[146]

## Data Availability

Not applicable.

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
