# Peer review of "Next-Generation Nanomedicine Approaches for the Management of Retinal Diseases"

_pharmaceutics, 2023, doi:10.3390/pharmaceutics15072005_

Round 1
Reviewer 1 Report
This review outlines the accessibility and constraints of existing therapeutic techniques while emphasizing understandings regarding the progress of forthcoming approaches in handling retinal conditions through the utilization of cutting-edge nanomedicine.
The manuscript is solid and covers a very important topic in a broad and consistent way.
After reading it, no deficiencies or gaps are detected and therefore it is recommended to publish it in its current format.
Author Response
We sincerely thank the reviewers for their time and valuable comments to improvise the manuscript. The manuscript has been revised based on other reviewer’s comments and changes are marked in blue color text throughout the manuscript.
The authors would like to thank the reviewer for critically evaluating the manuscript and providing a positive feedback.
Reviewer 2 Report
The review article titled “Next-Gen Nanomedicine Approaches for the Management of 2 Retinal Disorders” is well-written and has good scientific information. They compiled a review and discussed the current treatment strategies and their limitations and also highlights insights into the advancement of future approaches by nanomedicine to manage retinal diseases. The manuscript may be considered after minor revision on the following points:
· Table 3: The authors provided a characterization of the nanoformulation in the application part, it can be placed separately with details for each case study. First Case study: The dialysis method is not suitable; the cited reference described the radical polymerisation protocol developed by Rahimi and colleagues.
· Tables 4 and 5: The application column contains the characterization data which can be placed in separate columns. The authors need to describe the specific advantage or limitation of the nanocarrier discussed in the review.
· The authors can incorporate patents and clinical trials related to nanocarriers for retinal disorders which are very important and always provide information about translational research.
· The author can provide the conclusion for the review article.
The authors need to recheck the language for grammar and continuity in some places.
Author Response
We sincerely thank the reviewers for their time and valuable comments to improvise the manuscript. A point-by-point response to the reviewers' comments has been appended below. The manuscript has been revised and changes are marked in blue color text throughout the manuscript.
Table 3: The authors provided a characterization of the nanoformulation in the application part, it can be placed separately with details for each case study. First Case study: The dialysis method is not suitable; the cited reference described the radical polymerisation protocol developed by Rahimi and colleagues.
The characterization of the nanoformulation in the application section has been rearranged and placed separately, as suggested.
First Case Study: We apologize for the mistake. The method of preparation has been rectified in this section. The word "dialysis" is replaced with "radical polymerization," according to the cited literature.
Tables 4 and 5: The application column contains the characterization data which can be placed in separate columns. The authors need to describe the specific advantage or limitation of the nanocarrier discussed in the review.
As suggested, the characterization data has been included in a separate column in all the tables (Table 3, 4 & 5). The specific advantage or limitations of the nanocarrier has been described in each section.
The authors can incorporate patents and clinical trials related to nanocarriers for retinal disorders which are very important and always provide information about translational research.
Thanks for the suggestion but we are not aware of any clinical trial conducted for retinal disorders using nanocarriers.
The author can provide the conclusion for the review article.
We have included a conclusion section in the manuscript.
The authors need to recheck the language for grammar and continuity in some places.
We have checked and modified throughout the manuscript for grammer and continuity.
Reviewer 3 Report
The review summarizes the current research results of nanomedicinal preparations for the treatment of retinal diseases. The topic can be considered current, and it emphasizes diseases affecting the retina, which are occurring more and more frequently as a result of the aging population.
In the first part of the manuscript, current therapeutics for retinal diseases are presented. After that, nanomedicines are detailed. In the last chapter of the review, nanomedicines are systematically summarized from different aspects (overcoming blood-ocular barriers, cell-specific deliveries, safety and efficacy, drug deliveries, gene therapy and precision nanomedicine).
My comments and suggestions would be the followings:
1. The grouping of nanomedicines is questionable, why are nanoparticles not included in nanodelivery?
2. "Method of the preparation" is recommended instead of "Method" in the tables.
3. It would be good to see the administration way in the tables (topical eye drops, or intravitreal injection, etc.).
4. Table 3: the first mentioned method is dialysis; this could be more precise, because it is not clear.
5. Chapter 4 is a very good summary, but it would be advisable to close the article with a short conclusion.
Author Response
We sincerely thank the reviewers for their time and valuable comments to improvise the manuscript. A point-by-point response to the reviewers' comments has been appended below. The manuscript has been revised, and the changes are marked in blue color text throughout the manuscript.
Response to the Reviewer 3:
The grouping of nanomedicines is questionable, why are nanoparticles not included in nanodelivery?
The metallic nanoparticles can act as therapeutics without additional therapeutic molecules. On the other hand, the nanodelivery systems are much broader and include polymeric nanoparticles, lipid nanoparticles, liposomes, micelles, dendrimers, nanoemulsions, and nanogels. These delivery systems primarily act as drug carriers to execute their therapeutic effects. Therefore, the nanoparticles and nanodelivery systems are grouped separately.
"Method of the preparation" is recommended instead of "Method" in the tables.
They are amended as recommended.
It would be good to see the administration way in the tables (topical eye drops, or intravitreal injection, etc.).
Thanks for the suggestion. We have included the route of administration in the Tables.
Table 3: the first mentioned method is dialysis; this could be more precise because it is not clear.
We apologize for the mistake. The word "dialysis" has been replaced with "radical polymerization" in Table 3.
Chapter 4 is a very good summary, but it would be advisable to close the article with a short conclusion.
We appreciate your suggestion. We have included a short conclusion, as advised.